# Chlorine activated stacking fault removal mechanism in thin film CdTe solar cells: the missing piece

Peter Hatton [1,4], Michael J. Watts[2,4], Ali Abbas[3], John M. Walls[3,5], Roger Smith [1,5] & Pooja Goddard [2,5✉]

The conversion efficiency of as-deposited, CdTe solar cells is poor and typically less than 5%. A $CdCl_2$ activation treatment increases this to up to 22%. Studies have shown that stacking faults (SFs) are removed and the grain boundaries (GBs) are decorated with chlorine. Thus, SF removal and device efficiency are strongly correlated but whether this is direct or indirect has not been established. Here we explain the passivation responsible for the increase in efficiency but also crucially elucidate the associated SF removal mechanism. The effect of chlorine on a model system containing a SF and two GBs is investigated using density functional theory. The proposed SF removal mechanisms are feasible at the 400 °C treatment temperature. It is concluded that the efficiency increase is due to electronic effects in the GBs while SF removal is a by-product of the saturation of the GB with chlorine but is a key signal that sufficient chlorine is present for passivation to occur.

[1] Department of Mathematical Sciences, Loughborough University, Loughborough, Leicestershire, UK. [2] Department of Chemistry, Loughborough University, Loughborough, Leicestershire, UK. [3] Centre for Renewable Energy Systems Technology (CREST), Loughborough University, Loughborough, Leicestershire, UK. [4]These authors contributed equally: Peter Hatton, Michael J. Watts. [5]These authors jointly supervised this work: John M. Walls, Roger Smith, Pooja Goddard. ✉email: p.goddard@lboro.ac.uk

Electricity generated by photovoltaic modules is an important renewable source of power. The impact of solar electricity generation increases as device efficiencies improve and costs reduce. Although modules based on silicon absorbers dominate the current market, second-generation modules based on cadmium telluride (CdTe) are the most commercially successful thin film technology with over 25 GW already installed. Recent advances in thin film CdTe devices, notably the inclusion of selenium as an alloy at the front of the cell, has led to a sharp increase in module efficiency[1–3]. However, despite the impressive advances made, the conversion efficiency of thin film CdTe solar has the potential for further significant improvement because performance is still well short of the Shockley-Queisser theoretical limit, due mainly to the Voltage deficit. Further enhancement in device efficiency depends on an improved understanding of how these polycrystalline materials work and in particular how the detailed microstructure affects the electrical performance.

As-deposited thin film CdTe solar cells have a poor conversion efficiency typically <5% unless deposited at unusually high temperatures. Here, we show a cross-sectional image of an as-deposited device using Transmission Electron Microscopy (TEM), Fig. 1a. The CdTe absorber was deposited by close spaced sublimation (CSS). A notable feature of this image is the high density of stacking faults that appear as parallel lines the grain marked 'A'[4]. The stacking faults terminate at grain boundaries on either side of the grain and can be seen both in Fig. 2 and in the high-resolution TEM image of Fig. 3a.

The conversion efficiency of CdTe solar cells is transformed by an activation process[4]. In this process, CdCl$_2$ is deposited on the surface of the device held at a high temperature typically 400–430 °C for about 20 min. This process is used universally in research laboratories and in CdTe solar module manufacturing. During the process, chlorine rapidly diffuses along the grain boundaries and accumulates both there and at the device junction. The traditional n type buffer layer has been cadmium sulphide (CdS) and in this material the chlorine also diffuses along CdS grain boundaries and then accumulates along the interface with the transparent conductor (Fluorine doped tin oxide (FTO)) as shown in Fig. 1c. It is not observed to penetrate the FTO[5]. The CdS buffer layer in recently improved devices has been replaced with wide-band-gap metal-oxides such as magnesium-doped zinc oxide (MZO) to reduce optical absorption[6,7]. The chlorine does not decorate the MZO grain boundaries to the same extent and the chlorine accumulates mainly at the interface with the CdSeTe absorber. Although the chlorine has been observed in the grain interiors, it is mostly tightly bound to the CdSeTe or CdTe grain boundaries as shown in the TEM/EDX/NanoSIMS image in Fig. 1b–d. This tight segregation has been confirmed in computational work which showed that both interstitial and substitutional chlorine atoms were 2 and 1 eV more stable at the boundaries, respectively, than in the bulk[8].

The chlorine activation process causes a number of changes to the microstructure and improves the electrical performance. However, recrystallisation is minimal and little grain growth is observed following CdCl$_2$ activation on CdTe deposited with high-temperature deposition processes such as CSS, which is

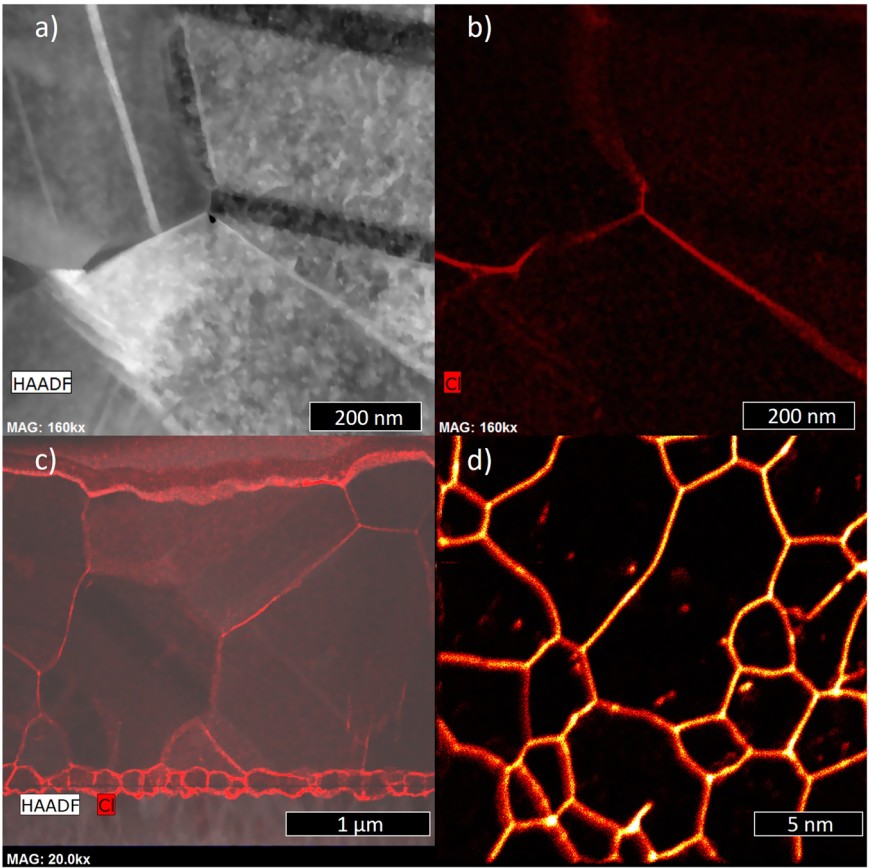

**Fig. 1 TEM/EDX/NanoSIMS images of Cl-treated CdTe. a** TEM image of activated CdTe showing grain boundary/interior structures. **b** Same region as in **a** with EDX mapping post-treatment CdTe with Cl shown in red; the Cl concentration is high in the grain boundaries compared to bulk. **c** Low-resolution TEM image of treated CdTe with EDX mapping of Cl atoms shown to have heavily segregated to the grain boundaries and interfaces. **d** $^{35}$Cl$^-$ NanoSIMS signal distribution on the back surface of the CdTe absorber showing Cl segregation at grain boundaries after CdCl$_2$ activation.

typically conducted at >500 °C compared to the lower temperature $CdCl_2$ process of 400–450 °C[9].

A previous study had been carried out to distinguish the effects of temperature and chlorine on the CdTe microstructure, which showed that while some reduction in the stacking fault density could be achieved with annealing only; the presence of chlorine was required for their complete and concerted elimination[10].

Further and conclusive evidence was obtained in another study in which previously $CdCl_2$ activated cells were annealed at temperatures above that used in the $CdCl_2$ treatment, i.e. >420 °C. TEM images showed that the stacking faults terminating at grain boundaries were restored and corresponding EDX maps showed that the chlorine had been removed from the grain boundaries[11]. This proved a direct link between the presence of stacking faults and the presence of chlorine in the grain boundaries.

Figure 2 shows cross-sectional TEM images of a CdTe layer, which has undergone partial $CdCl_2$ treatment. Stacking faults can be seen as parallel lines running throughout the grain interior. Adjacent grains appear to be in two distinct states of stacking

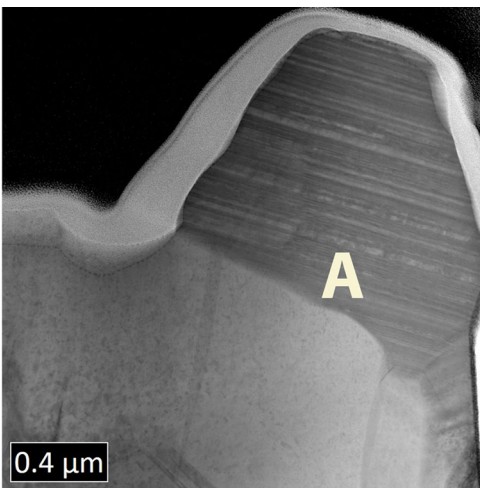

**Fig. 2 Low-resolution cross-sectional TEM of a CdTe layer.** A TEM image of a cross-section of thin film photovoltaic device after a partial activation treatment with cadmium chloride. The grain marked A contains a high density of stacking faults terminating at a grain boundary and at a free surface. Stacking faults have been completely removed in adjacent grains. Stacking fault removal in individual grains is concerted and associated with the density of chlorine located in the grain boundaries.

fault removal; either a high density of stacking fault layers can be seen or the grain is clean of these planar defects. These images suggest a stacking fault removal mechanism throughout a grain interior starting at adjacent grain boundaries where there is a local build up of chlorine. This implies that this build up of chlorine at an adjacent grain boundary is key to the removal of stacking faults in each individual grain where the resultant atomic structure can be seen in Fig. 3b after the high-temperature chlorine treatment.

The atomic scale mechanisms responsible for grain boundary passivation and stacking fault removal caused by the cadmium chloride treatment are still poorly understood. A DFT study of the electrical properties of stacking faults in CdTe showed that a range of intrinsic and extrinsic stacking fault arrangements are possible and that the high energy types, such as the polytype fault, would act as hole traps[12]. However high-resolution TEM experiments have shown that the stacking faults in as-deposited CdTe are predominantly tetrahedral faults[13] which DFT calculations have shown to be electrically benign[14] confirming that the removal of stacking faults is not directly responsible for the efficiency increase.

The most common tetrahedral fault occurring in CdTe is termed an intrinsic tetrahedral stacking fault[13–15]. A schematic for this type of fault is shown in Fig. 4. This fault is usually described as having two adjacent structure reflections and results in a local wurtzite structure in the otherwise zinc-blende lattice. It has been estimated from TEM images that 48% of layers in as-deposited CdTe have an intrinsic tetrahedral stacking fault structure with the majority of these terminating at grain boundaries[13].

The removal of such stacking faults during the chlorine treatment does not contribute to the efficiency increase of the cell[14]. Polytype stacking faults are shown to be detrimental to cell performance but these are not likely to form in significant numbers, due to having $\simeq$30 times the defect energy compared to other stacking fault types[12,14]. With few observations of polytype faults[13], the stacking fault removal process via the chlorine treatment is likely to be a mechanical by-product of a separate effect caused by the introduction of chlorine, which improves the efficiency.

It has recently been shown that grain boundaries in as-deposited CdTe create cross-boundary Te–Te interactions, which cause electronic defects in these regions[8]. These defects can increase electronic recombination causing the low efficiency of the as-deposited cell. This work also showed that if a single

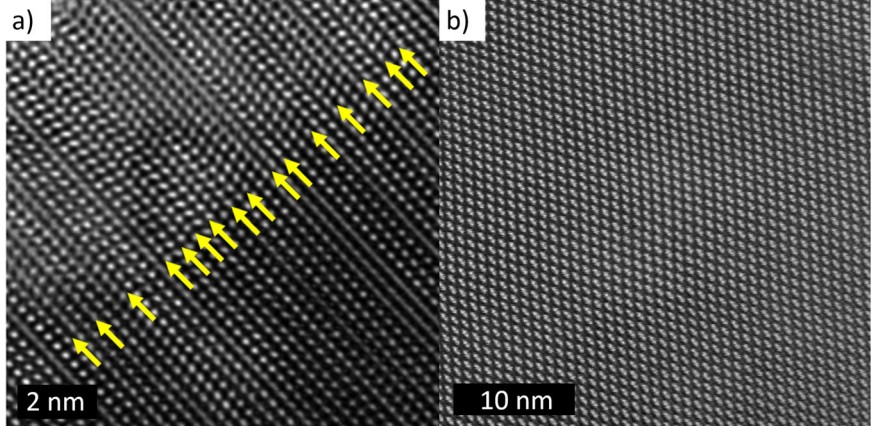

**Fig. 3 High-resolution TEM images of CdTe.** HRTEM images of CdTe (**a**) before and (**b**) after $CdCl_2$ treatment showing the existence of a high density of tetrahedral stacking faults (depicted by the yellow arrows) in as-deposited CdTe which are removed during the high-temperature chlorine treatment leaving a purely zinc-blende CdTe structure.

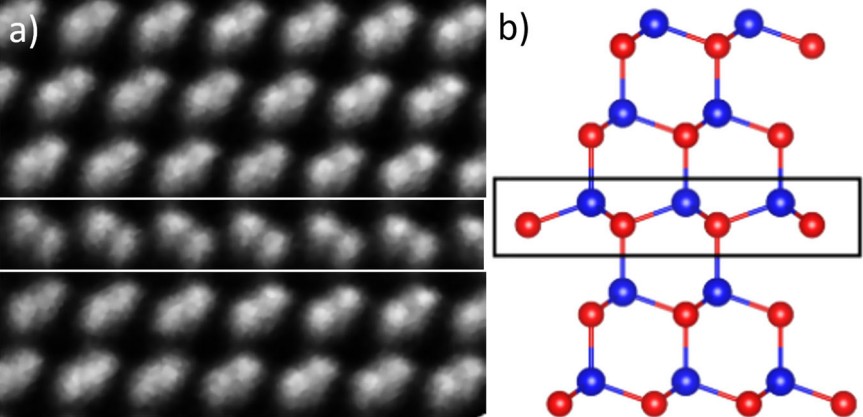

**Fig. 4 Intrinsic tetrahedral stacking fault. a** HRTEM image showing an Intrinsic Tetrahedral stacking fault[13]. **b** Schematic of this type of fault. Cd: Blue, Te: Red.

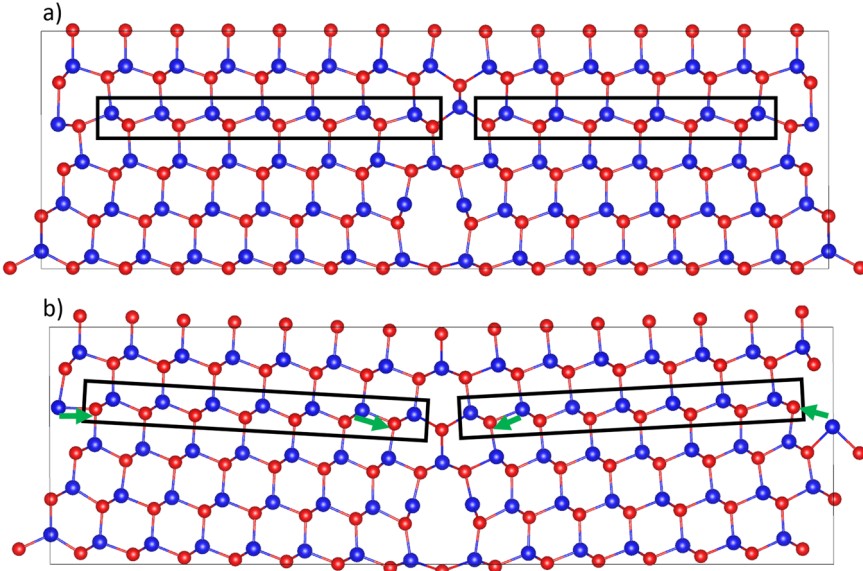

**Fig. 5 Clean Σ3 grain boundary structure. a** Stacking fault structure where the stacking fault layer between two grain boundaries is indicated by the rectangle. **b** The same structure with the stacking fault layer removed. The green arrows indicate the Te atom displacements required to transform to the structure shown in (**a**). Cd: Blue, Te: Red.

chlorine atom were to be present at either interstitial or substitutional sites in an isolated grain boundary structure both would result in defect passivation. If the same were true when many chlorine atoms were present, this would explain the efficiency increase of the CdTe solar cells after the chlorine treatment.

In this paper, we report on a detailed investigation into the relationship between conversion efficiency, grain boundary passivation with chlorine and the removal of planar defects. This work is a step change because it considers the effect of chlorine saturation in a model system consisting of a stacking fault terminating at two grain boundaries.

## Results and discussion

The model system consists of two intrinsic tetrahedral stacking faults between two Σ3 (112) grain boundaries, types commonly observed in TEM studies, one having a Cd- and the other having a Te-core. Before the addition of chlorine, 154 Cd and Te atoms were arranged in a system approximately $4.57 \times 62.08 \times 18.72$ Å with periodic boundary conditions applied in all directions. The system length was methodically increased to 62.08 Å when the

defect formation energy of a chlorine defect plateaued in the centre to give a bulk-like environment. The 4.57 Å thickness means effectively 1 atomic layer. Detailed studies of chlorine segregation to the grain boundaries can be found elsewhere[8]. The structure was relaxed using DFT to a local minimum energy configuration. The resulting system is shown schematically in Fig. 5a with the stacking fault layer indicated by the black rectangle.

Figure 5b shows the same system as in Fig. 5a but with the stacking fault removed. The relaxed structure shows a small shift in the angle of the layers in the grain interior, indicating strain in the system, but the structures of both grain boundaries are similar with only small changes. The result of this stacking fault removal is a system energy increase of 0.88 eV, implying that in the untreated cell the system including the stacking fault is more likely to form.

The energy barrier between the states shown in Fig. 5a, b has been found using a Nudged Elastic Band (NEB)[16,17] calculation, the results of which are shown in Fig. 6a. It must be noted that this is before chlorine treatment. The pathway between the two states is found to be through a cascade, which begins at both grain

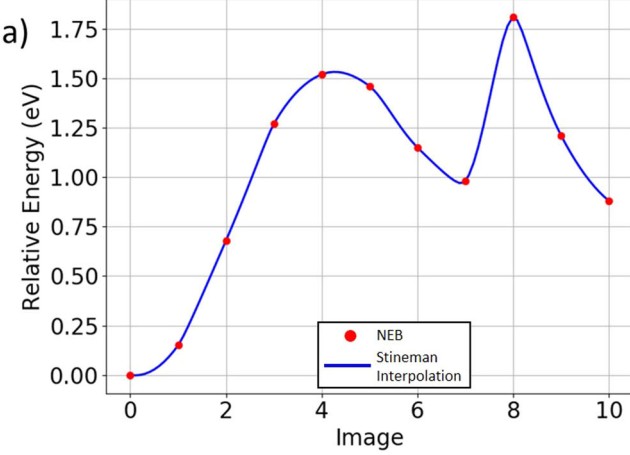

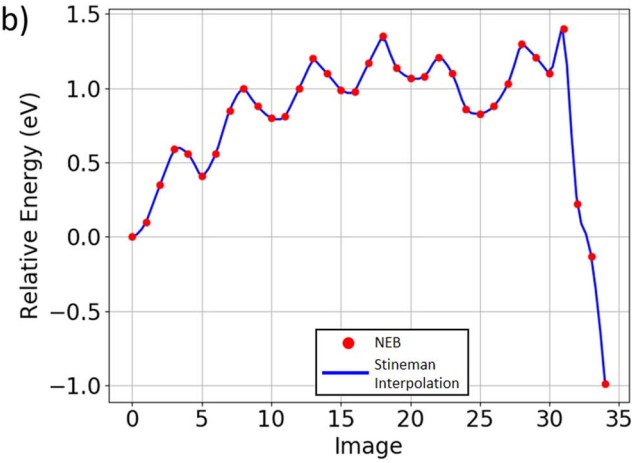

**Fig. 6 Nudged elastic band (NEB) calculations of stacking fault removal.** The energy pathway (**a**) between Fig. 5a, b. The intermediate minimum at image 7 represents the structure in which the stacking fault has been removed in the vicinity of the grain boundary but remains in the grain interior. **b** NEB calculation of the stacking fault removal process with 14 Cl. The change in energy along the pathway from Fig. 7a–f. Images 0–25 represent a Te cascade starting from the Cd-core and completely removing the stacking fault between the two cores. This mechanism can be seen in Fig. 7a–d. Images 25–34 represent the removal of the wrap-around stacking fault through the cell's periodic boundaries. The mechanism here is shown in Fig. 7d–f with an atomic mechanism similar to the 0 Cl case. Red points denote relative energies of individual energies, joined by a trend line in blue.

boundary cores simultaneously whereby a Te atom close to each boundary switches its bond to the adjacent Cd. The arrows in Fig. 5b indicate the Te diffusion direction, which initiates the stacking fault removal. The metastable site at image 7 in Fig. 6a represents the structure where the stacking fault has been removed in the vicinity of the grain boundary but in the middle of the grain interior, the stacking fault remains. A small barrier is required from this point to complete the transition and remove the entire stacking fault.

Since the transition from the system with the stacking fault to the system without involves a barrier of over 1.75 eV and an increase in energy of 0.88 eV, it is unlikely to occur in practice. Furthermore, the reverse barrier of 0.87 eV for re-introducing a stacking fault is much lower than to remove it. Experimentally, it has been shown that stacking faults return when Cl is removed through annealing as shown in Fig. 2. The transition mechanisms without the presence of chlorine suggests it is worth investigating

**Table 1 System energy change on stacking fault removal with varying amounts of Cl in the Σ3 structure of Fig. 5a, b.**

| Number of Cl atoms | Energy change |
| --- | --- |
| 0 | 0.88 eV |
| 9 | 0.86 eV |
| 12 | 0.27 eV |
| 13 | −0.46 eV |
| 14 | −1.02 eV |
| 15 | −2.56 eV |
| 16 | −2.18 eV |
| 18 | −1.54 eV |

if similar mechanisms exist when chlorine is present in the grain boundaries.

Before considering the stacking fault removal mechanisms themselves the relative stability of the structures shown in Fig. 5a, b is investigated as chlorine atoms are added to the grain boundaries and the energy difference between the relaxed structures, with and without the stacking fault, is determined.

Table 1 shows the difference between the systems with and without the stacking faults as the number of chlorine atoms in the grain boundary is increased. With each chlorine concentration, the structure is minimised with and without the stacking fault layer from a variety of starting chlorine positions. The chlorine configuration which resulted in the lowest energy is used and the energy change recorded. This data suggest that increasing the saturation of chlorine in the grain boundaries reduces the energy change on stacking fault removal. The switch over occurs with a total of 13 chlorine atoms in the system but the largest energy decrease is seen with 15 chlorine atoms (≃20% saturation) in the system, which has 6 chlorine atoms in the Cd-core and 9 chlorine atoms in the Te-core. At this point, the grain boundaries are saturated since the addition of further chlorine not only reduces the energy difference but the chlorine atoms also move into the grain interior during the relaxation process.

The determination of a pathway between the two states with and without stacking faults is more complex when chlorine was present compared to the case when it was absent. The procedure involved determining intermediate metastable states as suggested in the case without the chlorine and stitching these together piece by piece using the NEB method. This procedure was carried out for the system containing 14 chlorine atoms, the NEB calculation for which is shown in Fig. 6b with an overall barrier of 1.4 eV, easily overcome at 400 °C.

A full outline of the atomic mechanism of chlorine activated stacking fault removal is given in Fig. 7. The structure in Fig. 7a is identical to Fig. 5a but Cl has been added into interstitial and substitutional sites, which have previously been found to be important sites for defect passivation[8]. This resulted in two interstitial Cl atoms in the Cd-core and one substitutional Cl in the Te-core. A further 11 Cl atoms were placed interstitially in either the Cd- or Te-core. Figure 7 shows that after relaxation, some Te atoms in the grain boundary have been pushed to interstitial sites.

Images before and after the stacking fault removal with 14 chlorine atoms can be seen in Fig. 7a, f, respectively. Large reconstructions have occurred in both cores compared to the clean structures. The grain interiors remain chlorine-free and bulk-like but exhibit the same slight tilting seen with stacking fault removal without chlorine. It is worth pointing out that because of the periodic boundaries, the structure shown consists of two stacking faults, one between the grain boundary structures and one that wraps around through the periodic boundaries, these are both highlighted in the black rectangles of Fig. 7a.

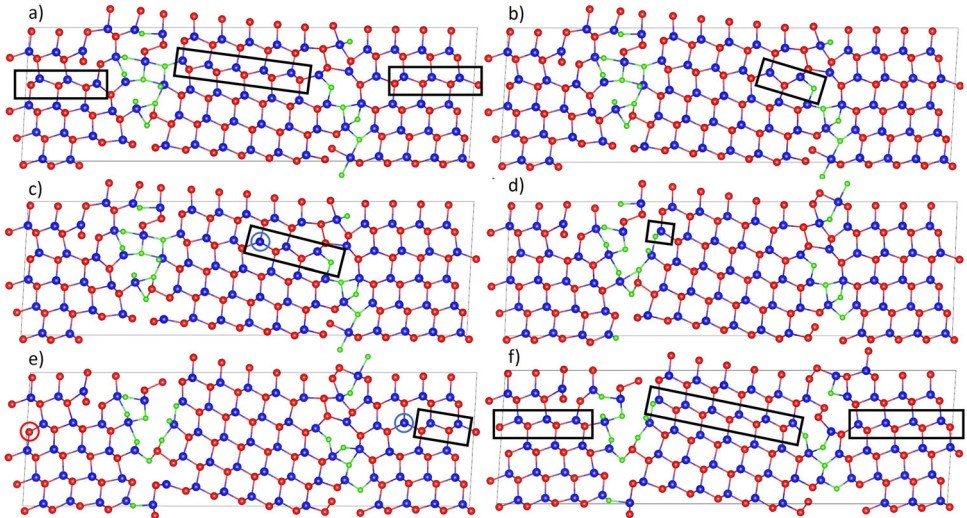

**Fig. 7 Atomic mechanism of stacking fault removal with Cl saturated grain boundaries.** The energy profile for this mechanism is given in Fig. 6b. **a** (Image 0) Lowest energy stacking fault structure with 14 Cl; the stacking fault layer is indicated by the rectangles. **b** (Image 5) The removal is initiated by a Te atom moving to an interstitial position with a 0.6 eV barrier shown in the rectangle. **c** (Image 10) The Te cascade has advanced by another step into the grain interior with a 0.6 eV energy barrier. **d** (Image 25) The Te cascade has completed the stacking fault removal between the grain boundaries with energy barriers of ≃0.4 eV. **e** (Image 30) The remaining wrap-around stacking fault is removed in the vicinity of both grain boundary cores simultaneously with 0.6 eV barrier. **f** (Image 34) The stacking fault is fully removed resulting in 2.1 eV energy decrease from previous metastable state and a 1 eV energy decrease from the original faulted structure. Cd: Blue, Te: Red, Cl: Green.

Different states along the pathway are also shown in Fig. 7. The initiation of the stacking fault removal occurs in Fig. 7b, the Cl in the rectangle acts as a Te substitution forming a Cl–Cd bond and a Te atom close to the Cd-core breaks its bond with an adjacent Cd and moves to an interstitial position highlighted in the black rectangle. The corresponding energies are shown in images 0–5 of Fig. 6b. Note that in the figure a Cl atom has replaced a Te atom where the SF meets the GB which initiates the SF removal process. This initial arrangement of the Cl atom occurred naturally during the minimisation when 14 Cl are added to the GB.

From this structure, a Te cascade is initiated propagating into the grain interior where each Te atom in the stacking fault layer switches the Cd it is bonded to. The result of the first of these movements can be seen in Fig. 7c. The stacking fault has only been removed in the region indicated by the rectangle but remains in the rest of the grain interior. Remarkably, this structure is similar to one that has been imaged experimentally by Chen Li et al.[18] finding that mid-grain dislocations create a mid-grain stacking fault terminating at Cd/Te dislocations. However, they were not able to elucidate the whole removal mechanism from this image alone nor do they have information on the saturation of chlorine that triggers this removal mechanism. In our example, the role of the Te dislocation is played by the Te-core grain boundary and a Cd dislocation has been created within the grain, indicated by the blue circle.

Subsequent barriers from image 15 to image 25 represent the cascading removal of the stacking fault via Te diffusing along the stacking fault layer. This mechanism has also been imaged by Chen Li et al.[18,19], showing the Cd dislocation approaching the Te dislocation and therefore the shortening of the stacking fault structure via Te diffusion until the dislocations annihilate creating a clean structure. Figure 7d shows the result of this cascading Te mechanism, each transition having a small ≃0.4 eV energy barrier. When the cascade reaches the middle of the grain interior the system energy plateaus. The process proceeds along the stacking fault layer until the inner stacking fault between the grain boundaries is removed. This occurs at image 25 of Fig. 6b

with the structure shown in Fig. 7d. The Cl indicated in the rectangle has substituted a Te and formed a Cl–Cd bond, which latches the resulting structure.

At image 25, the wrap-around stacking fault still remains but is subsequently removed by a similar Te cascade process that begins at the Cd and Te cores simultaneously. The metastable state at image 30, Fig. 7e, represents the wrap-around stacking fault having been removed in the vicinity of the grain boundaries but remaining in the grain interior further from the boundaries, indicated by the rectangle, similar to the mechanism with no chlorine. This structure creates a Cd- and Te- dislocation core shown in blue and red circles, respectively. This structure again is similar to one identified experimentally by Chen Li et al.[18,19] who saw this structure annihilate via the same Te cascade mechanism under the effect of the scanning beam used for atomic imaging, but not due to chlorine saturation. From the structure in 7e we estimate that the overall barrier required to complete this annihilation is 0.3 eV, shown in Fig. 6b images 30–34, which would be easily overcome at room temperature. The resultant structure can be seen in 7f.

The pathway suggested above will allow for complete removal of the stacking fault in the system and clearly other pathways may be possible but due to large structural shifts in the model and the computational expense involved in searching for these pathways this has not been considered further.

In recent experimental and modelling studies, chlorine-free CdTe grain boundaries, without stacking faults in the adjacent grain interior, have been shown to contain electronic defects, which increase electronic recombination due to mid-band-gap defects[20,21].

Previous work has focussed on attempting to passivate these defects by doping Te grain boundary sites with substitutional Cl. However, it has been recently shown that Cl interstitials will also be present at the grain boundaries through segregation calculations and are necessary to fully passivate electronic defects in these regions[8]. Therefore, this work shows the passivation of these defects through a combination of interstitial and substitutional Cl at the grain boundaries.

These calculations were carried out in a cell containing 136 Atoms in a system $9.14 \times 46.3 \times 11.25$ Å, i.e. double the thickness of our model. The passivation required three carefully placed chlorine atoms, aimed at blocking cross-boundary interactions between Te atoms, to fully passivate these defects. This suggests that a similar passivation effect might occur with grain boundaries that are saturated with chlorine.

The Density of States (DOS) plot of the stacking fault structure shown in Fig. 5 has been calculated using the hybrid functional HSE06 and is shown in Fig. 8a. Several defects are present in the band gap of the structure. The most damaging to cell efficiency would be the defect furthest from the Valence Band Maximum (VBM) between −0.4 and 0 eV, which is most likely to undergo Shockley-Reed-Hall recombination[22]. There are also defects close to the VBM, which are responsible for the flat region extending from the VBM at $\simeq -0.6$ eV. These defects in the structure with

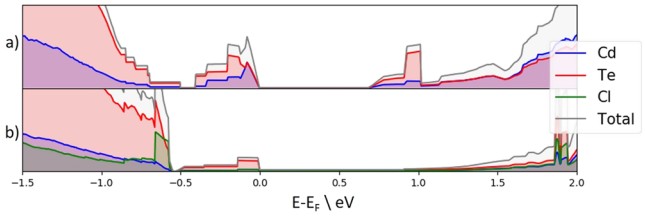

**Fig. 8 Normalised DOS grain boundary structure with varying Cl concentration.** Normalised DOS plot of **a** the clean Σ3 (112) grain boundary structure containing a stacking fault layer and **b** Σ3 (112) grain boundary structure containing 14 Cl without a stacking fault. A detailed study of the chlorine passivation and DOS plots of the clean GB with and without Cl stacking faults can be found in reference[8]. Comparing the DOS plots, it is clear that there is minimal effect from the stacking faults, especially in the mid-gap DOS.

the stacking fault are similar in nature to the defects found without the stacking fault layer and no Cl present[8] with small differences seen in the intensity of some defects most likely due to the thinner model.

Figure 9 shows the physical locations of the electronic defects, concentrated as expected, at the grain boundaries. The defect in Fig. 9a is responsible for the flat region extending from the VBM at $\simeq -0.6$ eV and is caused by the cross-boundary Te–Te interaction in the Cd-core. Since the stacking fault layer is above the Cd-core structure, the stacking of this layer does not impact the structure of the Cd-core. It is thus expected that this defect will be present regardless of the stacking fault existence. An identical defect was also found in a structure without the stacking fault[8]. Figure 9b, c is the defects responsible for the region between −0.4 and 0 eV in the DOS and both originate in the Te-core. The defect in Fig. 9b is concentrated on the cross-boundary Te–Te interaction at the top of the Te-core and this defect is accompanied by the Te at the bottom of the Te-core in Fig. 9c which, while being correctly coordinated, has non-standard bond angles with two adjacent Cd atoms. The physical location of these Te-core defects are different to those found without the stacking fault layer[8], but the overall conclusion of cross-boundary interactions invoking electronic defects at the grain boundaries, which cause defects in the band gap of the cell is the same.

The DOS plot for the model system with 14 chlorine atoms in the grain boundary and with the stacking fault removed is shown in Fig. 8b. This shows clear evidence of the chlorine passivation. The flat defect extending from the VBM due to cross-boundary Te–Te interaction in the Cd-core has been removed and there is a significant reduction in the intensity of the main mid-gap defect but with some defect remaining. There is also a some reduction of defect states around the CBM. Analysis shows that all identified defects in Fig. 9 have been removed and the remaining portion of the mid-gap defect is the result of Te noise throughout the structure related to the single atomic layer thickness necessary for

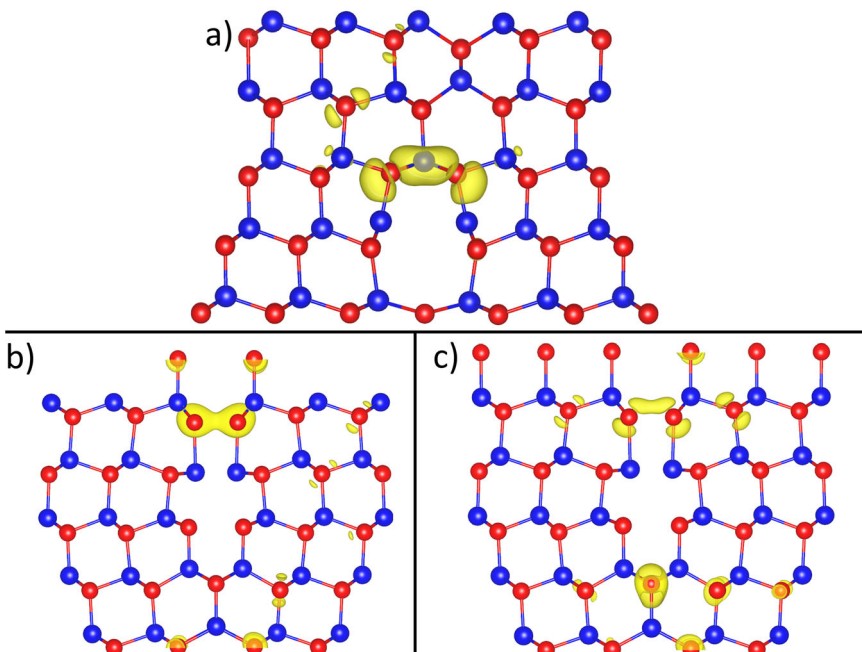

**Fig. 9 Partial charges at the grain boundaries with a stacking fault layer present. a** cross-boundary Te–Te interaction in the Cd-core responsible for the defect level at $\simeq -0.6$ eV in Fig. 8. **b** cross-boundary Te–Te interaction in the Te-core partially responsible for the defect level between −0.6 and 0 eV in Fig. 8. **c** Te defect caused by a non-standard Cd bond angle at the bottom of the Te-core partially responsible for the defect level between −0.6 and 0 eV in Fig. 8. Cd: Blue, Te: Red, yellow: charge isosurface.

the computationally expensive NEB calculations. This is in line with our previous work showing full passivation when thicker cells are used[8].

In conclusion, we report in this paper for the first time the full mechanism by which stacking fault removal occurs when CdTe solar cells are treated with chlorine. This occurs through a Te cascade mechanism that is triggered when the grain boundaries are saturated by chlorine. The process is demonstrated experimentally through high-resolution electron microscopy and nanoSIMS images, which clearly show the grain boundaries decorated with chlorine after the stacking fault removal. It is further shown that the grain boundaries saturated by both interstitial and substitutional chlorine remove defects in the band gap. Therefore concluding that the stacking fault removal is a by-product of the chlorine treatment and that the improved cell efficiency observed after the treatment is due to electronic defect suppression in the grain boundaries, solving the mystery of why and how chlorine is so effective in improving the CdTe cell efficiency. The removal of stacking faults is a useful but indirect, indicator that chlorine in the grain boundary has reached a sufficient concentration to passivate otherwise harmful defects. The fact that cell efficiency is always poor when stacking faults are present, means that the SF removal is a key signal that there is sufficient chlorine in the grain boundaries for passivation to occur.

## Methods

To keep the structures size to a minimum for NEB and minimisation operations, only a single CdTe unit thickness is used for the cell depth. The system size measures $4.58 \times 61.97 \times 18.73$ Å with a grain width of 31 Å and contains 154 atoms. The cell length was systematically increased until the defect formation energy of chlorine plateaued in the centre to give a bulk-like environment for the defect. A $4 \times 1 \times 2$ k-point grid is used with a 300 eV plane wave basis set cut-off for all DFT simulations of this cell, which were done using VASP[23]. NEB calculations used the modified solid state climbing image NEB[17,24,25] alongside the double-nudging method introduced by Wales et al.[26]. The size of the system was chosen so as to contain enough atoms that the effect of chlorine saturation in the grain boundaries and stacking fault removal mechanisms could be observed, without the necessity to use excessive amounts of computational resources. Structures were relaxed until the forces on all atoms converged to <0.01 eV/Å. DFT calculations were carried out using the PBEsol functional, which has been used previously for CdTe systems[8,14,27]. To study chlorine's effect on grain boundary defect passivation, hybrid DFT level theory was used with static calculations in a $4 \times 1 \times 2$ k-point grid using the HSE06 hybrid functional.

## Data availability

The input data used in this study have been deposited in the Loughborough University's figshare database under accession code[28] [https://doi.org/10.17028/rd.lboro.14842518].

## Code availability

The VASP 5.4.4 code[23] was used for the calculations. The script for the plotting of density of states in Fig. 8 is included with the data at https://doi.org/10.17028/rd.lboro.14842518[28].

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

## Acknowledgements

EPSRC Studentship 1801035; EPSRC Grant Nos. EP/P020232/1, EP/L000202, EP/R029431, EP/P020194.

## Author contributions

P.H. and M.J.W. ran the simulations and drafted the manuscript. M.J.W., P.G. and R.S. conceptualised the initial stacking fault removal mechanism. A.A., T.F. and J.M.W. performed all experimental works presented and contributed to the manuscript.

## Competing interests

The authors declare no competing interests.
