## [Peer Review File · Nature Communications]

REVIEWER COMMENTS

Reviewer #1 (Remarks to the Author):

This is a very satisfying wrap-up of an ongoing issue concerning the stacking faults that are so common in these solar cell materials. The authors have done a very thorough job of observing and calculating the energetics of the chlorine uptake and the stacking fault removal, nicely disentangling the mechanism. They conclude that the stacking fault removal is essentially a byproduct of the chlorination and the improved properties result from the chlorine at the grain boundaries. This adds significantly to our understanding, resolving the issues, and does essentially tie up the whole story, and so I think it deserves publication. The only comment I would like to give is that the authors might consider citing another paper by Chen Li where the movie and image is better at showing the annihilation process and they also calculate that the energy cost is low. It is C. Li, Y. Y. Zhang, T. J. Pennycook, Y. Wu, A. R. Lupini, N. Paudel, S. T. Pantelides, Y. Yan, and S. J. Pennycook, "Column-by-column observation of dislocation motion in CdTe: Dynamic scanning transmission electron microscopy," *Applied Physics Letters*, 109, 143107–6 (2016).

Reviewer #2 (Remarks to the Author):

The paper is important for CdTe PV technology.

Recommend publication after major revision.

1. The phrase "whole story" in the title is misleading. The authors have looked at σ_3 grain boundary one of the simplest ones and no comments about what may happen with more complex boundaries.
2. The caption in Figure 1 says as-deposited but EDX shows chlorine in the grain boundaries. Doesn't this mean that it is CdCl₂ treated?
3. Figure 5 needs more explanation. Te displacement leads to stacking fault removal but how chlorine causes Te displacement needs another figure.
4. The concept of "wrap around" stacking fault needs explanation.
5. Chlorine is generally thought to induce recrystallization. Couldn't the stacking faults be removed during recrystallization and chlorine accumulates at the grain boundaries; i.e. the accumulation not causing stacking fault removal?

Reviewer #3 (Remarks to the Author):

The authors studied Cl passivation at the CdTe GBs and its relation to the stacking fault. They find that Cl doping at the GB helps the removal of stacking fault.

Previous studies have already shown that Cl accumulation at the σ_3 (112) GB can help remove deep defect levels and the common stacking fault is not harmful to the system. This is somewhat confirmed in this paper and not much is new. So, the Cl-induced removal of stacking fault is not important in this field, and there are also other mechanisms that Cl can help remove stacking fault with or without GBs.

There are also some misleading and unclear statements in the paper:

What is the position of the Cl, how many are on interstitial sites or substitutional sites? How are they determined. It's not clear by just looking at Figure 8.

For computational easiness, both Te-Core and Te-Core are included in the small unit cell. What happens if only Te core GBs exist? What's the effect of interaction between Cd-core and Te-core GBs?

What are the calculated electronics states, where is the defect levels introduced by the GBs AND Cl?

Please plot a DOS of the clean Sigma3 (112) grain boundary structure without a stacking fault layer. Is the existence of stacking fault important?

In Figure 9b, there are still many mid-gap defect states and they claim it is the result of Te 'noise' throughout the structure related to the single atomic layer thickness necessary for the computationally expensive NEB calculations. What is their evidence for this statement and what do they mean 'noise'? Please also see [Phys. Rev. Lett. 101, 155501 (2008)], which show Cl cannot fully passivate the GBs.

They used the PBEsol functional for their calculations which underestimate the band gap. They said "A study of chlorine's effects on grain boundaries at the hybrid DFT level of theory is required to confirm and improve understanding of passivation mechanisms". But it is not clear whether they did it or not.

Based on the above observation, I think the paper should be rejected by Nature Communication.

Response to Reviewers

The authors would like to thank all three reviewers for their positive comments. We have addressed their individual points of concern below.

Reviewer #1:

This is a very satisfying wrap-up of an ongoing issue concerning the stacking faults that are so common in these solar cell materials. The authors have done a very thorough job of observing and calculating the energetics of the chlorine uptake and the stacking fault removal, nicely disentangling the mechanism. They conclude that the stacking fault removal is essentially a by-product of the chlorination and the improved properties result from the chlorine at the grain boundaries. This adds significantly to our understanding, resolving the issues, and does essentially tie up the whole story, and so I think it deserves publication.

- The only comment I would like to give is that the authors might consider citing another paper by Chen Li where the movie and image is better at showing the annihilation process and they also calculate that the energy cost is low. It is C. Li, Y. Y. Zhang, T. J. Pennycook, Y. Wu, A. R. Lupini, N. Paudel, S. T. Pantelides, Y. Yan, and S. J. Pennycook, "Column-by-column observation of dislocation motion in CdTe: Dynamic scanning transmission electron microscopy," *Applied Physics Letters*, 109, 143107–6 (2016).

Response: The authors agree that this reference is suited to the discussion of stacking fault annihilation in the grain interior and have added this reference as [19].

Reviewer #2:

The paper is important for CdTe PV technology.

Recommend publication after major revision.

1. The phrase "whole story" in the title is misleading. The authors have looked at σ_3 grain boundary one of the simplest ones and no comments about what may happen with more complex boundaries.

Response: We have now changed the title to "the missing piece" instead. We focused on the σ_3 boundary due to the complex nature of the calculations. Since the mechanism for stacking fault removal is very structural, we would expect a similar mechanism to remove the stacking faults as long as the structural conditions were met. A note to emphasise this has been added to the discussion, highlighted in blue.

2. The caption in Figure 1 says as-deposited but EDX shows chlorine in the grain boundaries. Doesn't this mean that it is CdCl₂ treated?

Response: *Figure 1a is as-deposited. All other images are after CdCl₂ treatment with figure 1b being the same region as in (a).*

3. Figure 5 needs more explanation. Te displacement leads to stacking fault removal but how chlorine causes Te displacement needs another figure.

Response: *The arrangement of the Cl in the GB and the subsequent atomic rearrangements and replacement of a Te atom by Cl to initiate the SF removal process was done by minimising many different structures rather than by molecular dynamics. As a result, we do not know the exact mechanism by which the Cl replaces the Te atom to start the removal cascade.*

4. The concept of "wrap around" stacking fault needs explanation.

Response: *The wrap-around stacking fault incorrectly referenced figure 5 in the original submission which does not in fact contain a wrap-around SF. The description of the stacking fault has been moved to accompany figure 8 which does have a wrap-around stacking fault. The explanation reads "It is worth pointing out that because of the periodic boundaries, the structure shown really consists of two stacking faults, one between the grain boundary structures and one that wraps around through the periodic boundaries, these are both highlighted in the black rectangles of figure 8a)."*

5. Chlorine is generally thought to induce recrystallization. Couldn't the stacking faults be removed during recrystallization and chlorine accumulates at the grain boundaries; ie. the accumulation not causing stacking fault removal?

Response: *This is a good question and we thank the reviewer for bringing it to our attention. In CdTe deposited via close space sublimation, stacking faults are also removed during CdCl₂ treatments which do not undergo significant recrystallization. This implies that there is some recrystallisation-independent mechanism at play. A discussion to clarify this has been added to the paper which reads "The chlorine activation process causes a number of changes to the microstructure and improves the electrical performance. However, recrystallisation is minimal and little grain growth is observed following CdCl₂ activation on CdTe deposited with high temperature deposition processes such as CSS which is typically conducted at > 500 °C compared to the lower temperature CdCl₂ process of 400 °C – 450° C [9]. A previous study had been carried out to distinguish the effects of temperature and chlorine on the CdTe microstructure which showed that while some reduction in the stacking fault density could be achieved with annealing only; the presence of chlorine was required for their complete and concerted elimination [10]. Further and conclusive evidence was obtained in another study in which previously CdCl₂ activated cells were annealed at temperatures above that used in the CdCl₂ treatment, i.e. > 420 °C. TEM images showed that the stacking faults terminating at grain boundaries were restored and corresponding EDX maps showed that the chlorine had been removed from the grain boundaries [11]. This proved a direct link between the presence of stacking faults and the presence of chlorine in the grain boundaries"*

[10] 'Effect of varying process parameters on CdTe thin film device performance and its relationship to film microstructure' Amit Munshi, Ali Abbas, John Raguse, Kurt Barth, W.S. Sampath, J.M. Walls Conference record of the IEEE 40th Photovoltaic Specialist Conference (PVSC), 1643-1648 (2014)

[11] 'The Effect of Annealing Treatments on Close Spaced Sublimated Cadmium Telluride Thin Film Solar Cells' J.M. Walls, A. Abbas, G. D. West, J.W. Bowers, P.J.M. Isherwood, P. M. Kaminski, B. Maniscalco, W.S. Sampath K. L. Barth . MRS Online Proceedings Library 1493, 61–66 (2012). <https://doi.org/10.1557/opl.2012.1704>

[12] 'The effect of a post-activation annealing treatment on thin film CdTe device performance' A Abbas, D Swanson, A Munshi, KL Barth, WS Sampath, GD West, J.W. Bowers , P.M. Kaminski and J.M. Walls. Conference record of 42nd Photovoltaic Specialist Conference (PVSC), 1-6 (2015)

Reviewer #3:

The authors studied Cl passivation at the CdTe GBs and its relation to the stacking fault. They find that Cl doping at the GB helps the removal of stacking fault.

Previous studies have already shown that Cl accumulation at the Sigma3 (112) GB can help remove deep defect levels and the common stacking fault is not harmful to the system. This is somewhat confirmed in this paper and not much is new. So, the Cl-induced removal of unarmful stacking fault is not important in this field, and they are also other mechanism that Cl can help remove stacking fault with or without GBs.

Response: We are fully aware that the most common stacking faults are not electrically harmful as reported previously by ourselves and others ref {14} in the paper. Whereas the stacking faults themselves may not be electrically harmful, the correlation between stacking faults removal and improved cell efficiency has never previously been explained and is important to understand since it is a by product of the chlorine treatment.

There are also some misleading and unclear statements in the paper:

- What is the position of the Cl, how many are on interstitial sites or substitutional sites? How are they determined. It's not clear by just looking at Figure 8.

Response: The authors agree that the nature of these Cl atoms is an important concept so have added a few passages to indicate how the Cl was initially inserted and to point out the nature of some of the key Cl atoms in the stacking fault removal process. Unfortunately, due to the high level of disordering during relaxation of these boundaries it is impossible to classify which Cl are interstitials and which are substitutions as everything is completely mixed.

Text has been added to explain how the Cl were initially inserted "The structure in figure 8a) is identical to figure 5a) but Cl has been added in to interstitial and substitutional sites which have previously been found to be important sites for defect passivation [8]. This resulted in two

interstitial Cl atoms in the Cd-core and one substitutional Cl in the Te core. A further 11 Cl atoms were placed initially interstitially in either the Cd- or Te-core. Figure 8 shows that after relaxation, some Te atoms in the grain boundary have been pushed to interstitial sites"

Text added to indicate the nature of the Cl atom indicated in the rectangle of figure 8b) reads "The initiation of the stacking fault removal occurs in figure 8b) with the Cl in the rectangle acting as a Te substitution forming a Cl-Cd bond." Text added to indicate the nature of the Cl atom indicated in the rectangle of figure 8d) reads "The Cl indicated in the rectangle has substituted a Te and formed a Cl-Cd bond."

Further text to explain has also been added "Note that in the figure a Cl atom has replaced a Te atom where the SF meets the GB which initiates the SF removal process. This initial arrangement of the Cl atom occurred naturally during the minimisation when 14 Cl are added to the GB."

- For computational easiness, both Te-Core and Te-Core are included in the small unit cell. What happens if only Te core GBs exist? What's the effect of interaction between Cd-core and Te-core GBs?

Response: *Due to the required periodicity of the cell, single core models would not be possible in the study of electronic defects since two oppositely polarised Sigma 3 cores are required to generate periodicity.*

The length of the grain boundary simulation cell was chosen to simulate a bulk-like environment in the centre of the cell which was confirmed using the defect formation energy of a Cl defect in this region. This comment has been added to the paper and reads "The system length of 62.08 Å was methodically increased until the defect formation energy of a chlorine defect plateaued in the centre to give to give a bulk-like environment."

In terms of the effect of the interaction between the two cores we believe this will be negligible in this study compared with the action of Cl. A detailed study of the effect of the interacting cores was reported previously in ref {8}. Further to this, increasing the length of the simulation cell to decrease the effect of the potentially interacting cores would make the HSE06 electronic analysis not viable due to the number of atoms required.

Text to refer the reader to ref {8} have been added. "Detailed studies of chlorine segregation to the grain boundaries can be found in ref {8}."

- What are the calculated electronics states, where is the defect levels introduced by the GBs AND Cl?

Response: *The core hypothesis of the paper is the stacking fault removal mechanism. The detailed electronic structure details requested in this point are already published in our previous publication, ref {8}. However, the following text has been added to clarify this in the paper, "Previous work has focussed on attempting to passivate these defects by doping Te grain boundary sites with substitutional Cl. However, it has been recently shown that Cl*

interstitials will also be present at the grain boundaries through segregation calculations and are necessary to passivate fully electronic defects in these regions ref{8}. Therefore, this work includes the passivation of these defects through a combination of interstitial and substitutional Cl at the grain boundaries.”

- Please plot a DOS of the clean Sigma3 (112) grain boundary structure without a stacking fault layer. Is the existence of stacking fault important?

Response: *We could do this but this is already published in ref 8. To make the point clear we have added this information to the caption to Fig. 9. We also know from previous reports and this reviewer has also suggested in an earlier point that the stacking faults are electrically benign because as we see here the stacking faults have minimal effect towards the conduction band minimum and not the band gap or the valence bands which are more critical for electrical conductivity.*

Text added, “A detailed study of the chlorine passivation and DOS plots of the clean GB with and without Cl stacking faults can be found in reference [8]. Comparing the DOS plots, it is clear that there is minimal effect from the stacking faults, especially in the mid gap DOS”.

- In Figure 9b, there are still many mid-gap defect states and they claim it is the result of Te ‘noise’ throughout the structure related to the single atomic layer thickness necessary for the computationally expensive NEB calculations. What is their evidence for this statement and what do they mean ‘noise’? Please also see [Phys. Rev. Lett. 101, 155501 (2008)], which show Cl cannot fully passivate the GBs.

Response: *There is clear evidence in the charge density distribution that the states are coming from Te “noise” related to the single atomic layer thickness. This is further evidenced in our previous paper, ref {8} where we show clearly that thicker cells do not have these states present. In the PRL paper quoted by the referee, the Cl is only considered at substitutional sites or interstitial sites in the Cd core. We have clearly shown in ref {8} that the presence of both substitutional and interstitials at both grain boundaries cores causes full passivation of the GB in thicker cells.*

To clarify this, text has been added, “This is in line with our previous work showing full passivation when thicker cells are used ref{8}.”

- They used the PBEsol functional for their calculations which underestimate the band gap. They said “A study of chlorine's effects on grain boundaries at the hybrid DFT level of theory is required to confirm and improve understanding of passivation mechanisms”. But it is not clear whether they did it or not.

Response: *To clarify further, this text has been changed to read “To study chlorine's effect on grain boundary defect passivation, hybrid DFT level theory was used with static calculations in a 4 X 1 X 2 k-point grid using the HSE06 hybrid functional.”*

REVIEWERS' COMMENTS

Reviewer #1 (Remarks to the Author):

The authors have responded to all issues raised. The paper is ready for publication.

Reviewer #2 (Remarks to the Author):

I am satisfied with the corrections made by the author and recommend publication.

Reviewer #3 (Remarks to the Author):

The authors have tried to address my questions, but they didn't answer the key question, that is, if the common stacking fault is not harmful to the system, why the removal of the stacking fault is important, and how it will improve the performance of the cell?

They also cannot explain well how the calculations was performed, such as how the CI sites are determined, which makes the repeating of their calculation almost impossible.

Some technical issues also exist. It is of cause possible to do calculation that contains single GB by passivating the supercell containing a single GB with pseudohydrogen, etc.

Based on the above observation, I don't think the paper is suitable for publication in NC.